# Immune Checkpoint Inhibitors in Malignant Pleural Mesothelioma: A Systematic Review and Meta-Analysis

**DOI:** 10.3390/cancers14246063

**Published:** 2022-12-09

**Authors:** Maria Gemelli, Diego Luigi Cortinovis, Alice Baggi, Pierluigi di Mauro, Stefano Calza, Alfredo Berruti, Salvatore Grisanti, Matteo Rota

**Affiliations:** 1Medical Oncology Unit, Istituto di Ricovero e Cura a Carattere Scientifico (IRCCS) MultiMedica, 20099 Milan, Italy; 2Department of Medical Oncology-ASST-Monza Ospedale San Gerardo, 20900 Monza, Italy; 3Department of Medical and Surgical Specialties, Radiological Sciences and Public Health, University of Brescia, 25123 Brescia, Italy; 4Unit of Biostatistics and Bioinformatics, Department of Molecular and Translational Medicine, University of Brescia, 25121 Brescia, Italy; 5Department of Medical Oncology-ASST Spedali Civili di Brescia, 25123 Brescia, Italy

**Keywords:** pleural mesothelioma, immune checkpoint inhibitors, immunotherapy, anti PD-1/PD-L1, anti CTLA-4

## Abstract

**Simple Summary:**

Many clinical trials have investigated the role of Immune Checkpoint Inhibitors (ICIs) in pleural mesothelioma (PM), with contrasting results. We performed a systematic review and meta-analysis of clinical trials testing single-agent ICIs or combined treatments in PM patients. Combined ICI treatments have higher Progression Free Survival (PFS) and Overall Survival (OS) rates when compared with single agents but a similar Overall Response Rate (ORR) and a higher rate of adverse events (AEs). ICI efficacy was independent of the treatment line.

**Abstract:**

Many clinical trials have investigated the role of ICIs in PM, with contrasting results. We performed a systematic review and meta-analysis of clinical trials testing single-agent anti-Programmed Death -1 (PD-1)/Programmed Death-Ligand 1 (PD-L1), anti-Cytotoxic T-Lymphocyte Antigen 4 (CTLA-4) or combined treatment in PM patients, analyzing response and survival rate as well as safety data. We selected 17 studies including 2328 patients. Both OS and PFS rates were significantly higher with combined ICI treatments than with single agent anti-PD-1/PD-L1 (*p* < 0.001 and *p* = 0.006, respectively) or anti CTLA-4 (*p* < 0.001) treatments. ORR and DCR for all ICI treatments were 20% (95% CI 13–27%) and 56% (95% CI 45–67%), respectively, and they did not significantly differ between combined and single agent treatments (*p* = 0.088 and *p* = 0.058, respectively). The 12-month OS and 6-month PFS rates did not differ significantly (*p* = 0.0545 and *p* = 0.1464, respectively) among pre-treated or untreated patients. Combined ICI treatments had a significantly higher rate of Adverse Events (AEs) (*p* = 0.01). PD-L1-positive patients had a higher probability of response and survival. In conclusion, combined ICI treatments have higher efficacy than single agents but are limited by higher toxicity. Efficacy was independent of treatment line, so a customized sequential strategy should still be speculated. PD-L1 expression could influence response to ICIs; however, reliable biomarkers are warranted.

## 1. Introduction

Pleural mesothelioma (PM) is a remarkably aggressive disease associated with asbestos exposure [1]. Its incidence has increased in recent decades in industrialized countries due to the widespread use of asbestos and the time delay in tumor development following asbestos exposure, typically 30–50 years [2]. Three histologic variants with different prognoses have been identified: epithelioid histology with the longest survival (12–27 months), sarcomatoid with the shortest (7–18 months), and biphasic with intermediate outcomes (8–21 months) [3,4]. Most patients have an inoperable disease at diagnosis and in this setting platinum agents plus pemetrexed, with or without bevacizumab, have been the only approved first-line treatment regimen from 2004 to October 2020 [2,5,6]. The therapeutic strategy in refractory patients has not yet been clearly established [7,8,9,10,11,12].

More recently, the combination of nivolumab and ipilimumab has been shown to have improved survival compared with platinum–pemetrexed chemotherapy in untreated patients, which was proven in the Checkmate 743 trial, establishing a new standard for the first-line treatment [13]. However, although the magnitude of benefit was clearly superior for Immune Checkpoint Inhibitor (ICI) combinations in non-epithelioid histologies, it was less evident in the epithelioid subtype. Furthermore, combination immunotherapy was hampered by a higher rate of serious Adverse Events (AEs), which makes it unsuitable for frail and elderly patients [13]. Thus, whether ICI combinations should be the preferred choice in these patients in the first-line setting is still a matter of debate. Some authors have questioned the real survival benefit of the combination immunotherapies by performing an indirect comparison with the standard platinum–pemetrexed +/− bevacizumab [14,15]. In addition, retrospective analysis of real-world data did not suggest a clear survival benefit for immunotherapy in PM patients [16]. In the meantime, results of a phase III randomized clinical trial with chemotherapy–immunotherapy combinations are awaited [17,18].

Vinorelbine or gemcitabine are widely used in pre-treated patients but have only modest activity, with the Overall Response Rate (ORR) ranging from 0 to 2% [7]. More recently, in a phase II randomized VIM trial, vinorelbine altered the Progression Free Survival (PFS) rate compared with active supportive care (4.2 vs. 2.8 months, respectively); the primary endpoint was met even if the use of chemotherapy did not affect the Overall Survival (OS) [8]. On the contrary, a randomized phase II trial showed that the addition of ramucirumab, an anti-vascular endothelial growth factor receptor 2 (VEGFR-2) monoclonal antibody, improved OS compared with gemcitabine and placebo; however, this was without any improvement in the ORR [19]. The survival benefit of this combination was questioned by a possible selection bias, with an imbalance in subgroups regarding an Eastern Cooperative Oncology Group (ECOG) performance status of 0 and an age younger than 70 years, two well-known prognostic factors in PM, in favor of the ramucirumab group. Furthermore, many concerns have been raised about the choice of the control arm: gemcitabine alone might not be considered an optimal second-line control treatment, particularly in late (>6 months) progressors in lieu of a possible first-line drug rechallenge.

More generally, relapsed or refractory patients lack valid therapeutic alternatives; in a meta-analysis, Petrelli et al. reported an ORR of 8.63% and a median PFS and OS of 3.4 and 7.86 months, respectively, for second-line treatments [20].

Data on the use of ICIs in pre-treated patients are contrasting and suggest only mild efficacy [21]. Cytotoxic T-Lymphocyte Antigen 4 (CTLA-4) inhibitors provided only limited efficacy, whereas small trials with single agent pembrolizumab, nivolumab and avelumab showed a potential benefit for the use of Programmed Death-1(PD-1)/Programmed Death-Ligand 1 (PD-L1) blockade in the second-line treatment of PM, leading to the NCCN guidelines (2018) recommendation to use nivolumab ± ipilimumab or pembrolizumab as subsequent systemic therapy in patients with progressive disease after first-line chemotherapy [9,10].

Only two phase III randomized trials have been conducted in relapsed PM patients, which differ in control arms and primary endpoints. The phase III randomized CONFIRM trial showed a statistically significant survival benefit for nivolumab compared with placebo (HR 0.69, 95% CI 0.52–0.91; *p* = 0.0090). In the phase III PROMISE-meso trial, despite a higher ORR, pembrolizumab did not improve survival compared with single-agent chemotherapy; both median PFS and OS were numerically inferior [11,12]. Additionally, the use of placebo as the control arm in the CONFIRM trial raised many ethical concerns and might limit the clinical impact of the survival advantage observed in the trial [12]. For all of these reasons, no second-line treatment has been clearly established as standard and inclusion into clinical trials is currently the best option.

At present, the real clinical impact of immunotherapy in the management of PM is difficult to ascertain due to several factors, including the low number of patients included in early clinical trials, the different lines of treatment in which the ICIs are employed, and the different combinations of therapeutics used. For these reasons we decided to perform a meta-analysis of published data to assess the overall impact of ICIs in patients with PM.

## 2. Materials and Methods

This systematic review followed the Preferred Reporting Items for Systematic Reviews and Meta-Analyses (PRISMA) 2020 statement [22]. A protocol was created and registered on The International Prospective Register of Systematic Reviews (PROSPERO) website (Registration No.: CRD42021229532). Institutional review board permission was not necessary, as no individual patient-data was used for the meta-analysis.

### 2.1. Searching Strategies and Data Sources

Studies published in English between 2005 and 30 April 2021 were searched for among two databases: Medline/pubmed and EMBASE.

The following MeSH terms were applied: “Mesothelioma”, “Pleural Neoplasms”, “Immune Checkpoint Inhibitors”, and “Immunotherapy”. The complete literature research string is reported in the Appendix A. For the sake of completeness, references of retrieved relevant articles were reviewed to identify any other potential articles not identified through the electronic literature search.

### 2.2. Study Selection and Eligibility Criteria

Published phase I- II- or III randomized controlled trials (RCTS) assessing the efficacy of ICIs (anti PD-1/PD-L1) as monotherapy or in combination with chemotherapy or immunotherapy in patients with PM were included, as well as abstracts regarding immunotherapy and chemotherapy from international congresses (ASCO-ESMO-WLCC) on immunotherapy and chemotherapy. Reviews, comments, case reports, expert opinion, and animal studies were excluded.

### 2.3. Objectives of the Study

The main objective of this meta-analysis is to assess the efficacy of ICIs as single-agent or combination therapies in PM, comparing the effect of anti-CTLA-4 and anti-PD-1/PD-L1 agents and between single or combination agents in the first or second lines of therapy. The primary endpoints of the study are: OS at 12 months and PFS at 6 months. The secondary endpoints are: ORR, Disease Control Rate (DCR), and safety.

### 2.4. Data Extraction

Two independent investigators (A.B and P.dM) extracted all data from the eligible studies. A third investigator checked the data (M.R.). Any inconsistency was discussed within the group. Data collected from the eligible studies were study name, year of publication, treatment regimen and line, information on PD-1, histologic subtypes, patient characteristics (median age, sex, and ECOG PS), sample size, outcomes (OS, PFS, ORR, and DOR), adverse events, and measures of effect (HR, OR, 95%CI, and *p*-value) when available. If the HR and its 95% CI for OS or PFS were not reported, we calculated them according to the published data. OS and PFS survival curves were digitized through a semi-automatic freely available tool (WebPlotDigitizer) that allows extraction of OS and PFS estimates for each follow-up timepoint [23].

### 2.5. Statistical Analysis

As the majority of the included studies were single-arm, OS and PFS could not be summarized in terms of hazard ratio (HR). Therefore, OS and PFS were synthetized across studies as proportions of patients surviving at 6 and 12 months. For the same reason, ORR, DCR, and AEs were evaluated as proportions too. For the stratified analyses according to PD-L1 expression, OS and PFS were synthetized in terms of HR. If the eligible studies did not report HR estimates, we computed them from Kaplan–Meier curves using the methods described in Tierney et al. [24]. ORR and DCR according to PD-L1 expression were summarized as odds ratio (OR). As we anticipate heterogeneity among RCTs, study-specific estimates were summarized across studies by means of a random effects model: each study-specific estimate was weighted by the inverse of its variance plus an estimate of the between-study variance component τ^2^ estimated through the DerSimonian and Laird moment estimator [25]. Results were displayed in forest plots. Between-study heterogeneity was assessed through the Q test based on the χ^2^ statistics, whereas the I2 statistic was used to quantify the proportion of the total observed variability attributed to study heterogeneity. A leave-one-out sensitivity analysis was performed by iteratively removing 1 study at a time to assess whether the pooled estimate was influenced by any of the eligible studies. Publication bias was assessed by visual inspection of funnel plots for asymmetry and by means of the Egger’s test if the number of eligible studies was greater than 10.

### 2.6. RISK of BIAS (Quality) Assessment

The Cochrane risk of bias tool (RoB) was used to assess the risk of bias of eligible RCTs. Briefly, bias is assessed as a judgment (high, low, or unclear) for individual elements from five domains (selection, performance, attrition, reporting, and other). Two independent investigators assessed the quality of the studies and any divergences between them were resolved by discussion. Appendix A reports the risk of bias assessment.

## 3. Results

### 3.1. Literature Research

Four-hundred sixty-six studies were identified from databases and registers. After electronic searching, title/abstract screening, and full-text review, 14 published studies met our eligibility criteria. We also identified three studies from the websites associated with global conferences that were eligible for our analysis. At the end of the process, 17 trials were retained for analysis. The PRISMA 2020 flow diagram and detailed reasons for study exclusion are presented in Figure 1.

The 17 trials included 2328 patients [27,28,29,30,31,32,33,34,35,36,37,38,39,40,41]. Characteristics of the included trials and the outcomes of interest are reported in Table 1 and Table 2. Five of them are RCTs [11,12,13,32,37], three in pre-treated patients [11,12,32], one in naïve patients [13], and one enrolled naïve and pre-treated patients in two different cohorts [37]. Of the 17 trails, 3 were phase III [11,12,13], 12 were phase II [27,28,29,30,31,32,33,36,37,38,39,40,41], and 2 were phase I trials [34,35]. Two of the five RCTs compared the experimental treatment with an active treatment (chemotherapy) [11,13] and two with placebo [12,32]. One trial did not compare but randomly allocated patients to receive combination (i.e., ipilimumab and nivolumab) versus single-agent (nivolumab) immunotherapy [37]. The primary endpoint was OS for three studies, PFS for three, ORR for nine studies, and DCR for four studies.

### 3.2. Systematic Review: First-Line Treatments

Single-agent immunotherapy has not been investigated in untreated PM patients. The combination of durvalumab and tremelimumab was tested in the open-label, non-randomised, phase II NIBIT-MESO-1 trial, both in untreated and pre-treated patients with unresectable PM [27,28]. The primary endpoint was ORR according to modified RECIST [27]. After a median follow-up of 19.2 months (IQR 13.8–20.5), 11 (28%) of 40 patients had a partial response (PR), with a median DOR of 16.1 months (IQR 11.5–20.5) [27]. Median PFS was 5.7 months (1.7–9.7) and mOS was 16.6 months (13.1–20.1). Baseline tumor PD-L1 expression did not correlate with response nor survival [27,28].

The multicenter open- label, randomized, phase III study, CheckMate 743, randomized 605 untreated, unresectable PM patients to receive nivolumab plus ipilimumab for up to 2 years or platinum plus pemetrexed chemotherapy [13]. The primary endpoint was OS. At the median follow-up of 29.7 months (IQR 26.7–32.9), nivolumab plus ipilimumab significantly prolonged OS versus chemotherapy (18.1 months [95% CI 16.8–21.4] vs. 14.1 months [12.4–16.2]; HR 0.74 [96.6% CI 0.60–0.91]; *p* = 0.0020). OS favored nivolumab plus ipilimumab across most pre-specified subgroups, except in patients aged 75 years and older (*n* = 157). OS was improved with nivolumab plus ipilimumab versus chemotherapy regardless of histology; however, the magnitude of benefit was higher in non-epithelioid (HR 0.46 [95% CI 0.31–0.68]) than the epithelioid subtype (0.86 [0.69–1.08]) [13]. Furthermore, the OS benefit of nivolumab plus ipilimumab was greater in patients with a PD-L1 expression of 1% or higher (HR 0.69 [95% CI 0.55–0.87]) than in PD-L1-negative patients (0.94 [0.62–1.40] [13]. The median PFS was similar between treatment groups (HR 1.00 [95% CI 0.82–1.21]), as was the ORR (40% in the nivolumab plus ipilimumab group vs. 43% in the chemotherapy group) [13]. Grade 3–4 treatment-related adverse events were reported in 91 (30%) of 300 patients treated with nivolumab plus ipilimumab and 91 (32%) of 284 treated with chemotherapy. Three (1%) treatment-related deaths occurred in the nivolumab plus ipilimumab group (pneumonitis, encephalitis, and heart failure) and one (<1%) in the chemotherapy group (myelosuppression) [13].

Durvalumab has been investigated in combination with cisplatin and pemetrexed in the multicenter phase II DREAM single-arm trial [29]. Primary endpoint was PFS at 6 months from enrollment. At the median follow-up of 28.2 months, 31 (57%; 95% CI 44–70) of 54 patients were alive and progression free at 6 months; the mPFS was 6·9 months (95% CI 5.5–9.0). The ORR was 48% (95% CI 35–61), and the mOS was 18·4 months (95% CI 13.1–24.8) with a 12-month overall survival of 65% (54–79) and a 24-month overall survival of 37% (26–52) [29]. On this basis, the phase III trial is ongoing [NCT04334759].

### 3.3. Systematic Review: Second-Line Treatments

The anti-CTLA-4 treatment, tremelimumab, has been investigated as single agent in two phase II trials in pre-treated mesothelioma patients. In the larger one, the DETERMINE trial, 571 pre-treated mesothelioma patients were randomized to receive tremelimumab or placebo; the study did not meet its primary OS endpoint but showed a statistically significant, although not clinically relevant, PFS benefit in favor of tremelimumab (2.8 vs. 2.7 months, *p* = 0.03) [32].

Preliminary data on nivolumab efficacy in pre-treated mesothelioma patients was derived from two phase II single-arm trials, the Dutch NivoMes and the Japanese MERIT trials, both showing encouraging data on OS and PFS (Table 2) [38,39]. The phase III CONFIRM trial randomized 332 patients with MPM to receive nivolumab or placebo, with nivolumab outperforming placebo in both its primary endpoints, PFS (3.0 vs. 1.8 months, *p* < 0.001) and OS (10.2 vs. 6.9 months, *p* = 0.009) [12].

Among the RCTs, the phase II IFCT-1501 MAPS 2 study, which did not formally compare but randomly allocated patients to either an immunotherapy combination of anti-CTLA-4 ipilimumab and anti-PD1 nivolumab or nivolumab alone and had DCR at 12 weeks as the primary endpoint, showed a numerically longer PFS (5.6 vs. 4.0 months) and OS (15.9 vs. 11.9 months) in favor of the immunotherapy combination over the nivolumab monotherapy [37].

The combination of nivolumab and ipilimumab has also been investigated in the phase II single-center, single-arm, INITIATE trial [33]. DCR at 12 weeks, the primary endpoint, was 68% (95% CI 50–83), whereas OS at 12 months was 64% (Table 2). [33] The small number of non-epithelioid patients did not allow a meaningful comparison between histological subtypes. No stratification for PD-L1 was performed. AEs were reported in 33 (94%) patients, which were most commonly infusion-related reactions, skin disorders, and fatigue. Grade 3 AEs were reported in 12 (34%) of 35 patients [33].

Pembrolizumab was first tested in the phase IB multi-cohort KEYNOTE-028 trial in patients with PD-L1-positive solid tumors [35]. In the PM cohort, 25 patients received pembrolizumab at 10 mg/kg q2w for up to 24 months. The primary endpoints were safety and ORR. At the median follow-up of 18.7 months, the ORR was 28% and the DCR was 76% [35]. On this basis, pembrolizumab was further investigated in a dedicated cohort of the phase II multi-cohort KEYNOTE-158 study independently of PD-L1 expression. [36]. ORR, the primary endpoint, was 8%, with 14.3 months of median DOR and 60% objective response ongoing at 12 months [36]. Desai et al. conducted a two-part, phase II, single-center, non-randomized trial [40]. Part A enrolled 35 PM patients to determine the ORR with pembrolizumab treatment and to find the optimal PD-L1 cut-off for positivity [40,42]. Part B was initiated when seven responses were reported in part A and was intended to use a biomarker enrichment strategy for PD-L1 [40,42]. However, no PD-L1 cut-off was defined in the first part, so the second part enrolled 30 patients irrespective of PD-L1 level. mRECIST criteria were used to assess the radiologic response. The ORR was 22% and the DCR was 63%. The median PFS and OS were 4.1 and 11.5 months, respectively [40,42]. PD-L1 expression ≥50% was associated with a higher RR and longer median PFS [42]. The PROMISE-meso was the only phase III randomized trial comparing pembrolizumab or chemotherapy (gemcitabine or vinorelbine) in PM patients who progressed to platinum-based chemotherapy [11]. PD-L1 expression analysis was exploratory. The primary endpoint was PFS by blinded independent central review. After a median follow-up of 11.8 months, the study did not meet its primary endpoint; median PFS with pembrolizumab was 2.5 months compared with 3.4 months for the control (HR 1.06, 95% CI: 0.73–1.53, *p* = 0.76 stratified by histological subtype) [11]. Nevertheless, the ORR was superior with pembrolizumab than with chemotherapy (22% vs. 6%, *p* = 0.004), whereas the median DOR was higher with chemotherapy (7.2 months vs. 4.6 months). The median OS was not statistically different between study arms. No differences were observed in median PFS or OS according to PD-L1 expression (TPS < 1% vs. >1%) [11].

Avelumab was evaluated in 53 pre-treated patients within the phase Ib JAVELIN trial who were stratified for histology and PD-L1 expression (cut-off for positivity of 5%) [34]. After a median follow-up of 24.8 months, the ORR was 9% (95% CI 3.1–20.7) and the DCR was 58% (*n* = 31), whereas the median PFS and OS were 4.1 (95% CI 1.4–6.2) and 10.7 (95% CI 6.4–20.2) months, respectively. The ORR was higher in PD-L1-high than in PD-L1-low/negative patients (*p* = 0.034), as was the mOS (20.2 vs. 10.2 months) CTLA [34].

The anti-PD-L1 agents atezolizumab and durvalumab have not been investigated as single agents in prospective trials in this setting.

### 3.4. Meta-Analysis: Overall Outcomes

Overall, the 12-month OS rate with ICI treatments is 53% (95% CI 44–61%). The pooled-estimate 12-month OS rate is significantly higher with combination treatments (66%; 95% CI 61–70%) than with single-agent anti-PD-1/PD-L1 (51%, 95% CI 45–57%; *p* < 0.0001) or anti CTLA-4 (40%, 95% CI 27–54%) (*p* < 0.0001) treatments (Figure 2A). The 6-month PFS rate with ICIs is 19% (95% CI 13–25%). The 6-month PFS rate is significantly higher with combination treatment (29%; 95% CI 24–33%) than with single-agent anti-PD-1/PD-L1 (16%, 95% CI 12–20%) (*p* = 0.006) or anti-CTLA-4 (5%; 95% CI 1–12) (*p* < 0.0001) treatments (Figure 2B). No studies have a significant influential effect. However, when considering studies grouped by agent, the DREAM trial [29] has an influential effect on 12-month OS, as do the DETERMINE [32] and CHECKMATE 743 trials [13]. The NIBIT-MESO-1 [27,28] and MAPS2 [37] trials exhibited an influential effect on 6-month PFS. Appendix A reports funnel plots for 12-month OS (Panel A), 6-month PFS (Panel B), ORR (Panel C), and DCR (Panel D). Evidence of asymmetry emerged for 6-month PFS (*p* = 0.001).

Taken together, ICIs show a 20% ORR (95% CI 13–27%): 21% for anti-PD-1/PD-L1 (95% CI 13–28%), 5% for anti-CTLA-4 (95% CI 2–9%), and 26% for the combination of anti-CTLA-4 and anti-PD-1/PD-L1 (95%CI 12–39%) (Figure 2C). The pooled-estimate DCR was 56% (95% CI 45–67%). Combination treatments have the highest DCR (64%, 95% CI 52–76%), followed by single-agent PD-1/PD-L1 (59%, 95% CI 47–71%), and single-agent anti-CTLA-4 (35%, 95% CI 21–48%) (Figure 2D).

Overall, no studies have a significant influential effect on the ORR and DCR pooled estimates. The ORR and DCR were not statistically significantly different between the combined or single-agent anti-PD-1/PD-L1 treatments (*p* = 0.088 and *p* = 0.058, respectively), whereas they were significantly higher when compared with anti-CTLA-4 (*p* < 0.0001). When considering studies grouped by agent, the DREAM trial [29] has an influential effect, as does the DETERMINE trial [32] and the MESOTREM—2008 [30] for ORR only and the MESOTREM-2012 trial [31] for DCR only. For the combination of anti-CTLA-4 and anti-PD-1/PD-L1, the NCT03075527 [41] and the MAPS2 trials [37] exhibited an influential effect on ORR and DCR, respectively.

PD-L1-positive patients have a 2.24 (95% CI, 1.27, 4.12) higher chance of a response compared with PD-L1-negative patients with anti-PD-1/PD-L1 or combination treatments. The OR was higher with single-agent anti-PD-1/PD-L1 (OR 2.49, 95% CI 1.06–5.84) than with combined anti-PD-1 and anti-CTLA-4 (OR 2.04, 95% CI, 0.92–4.54) (Appendix A).

Furthermore, PD-L1-positive patients have 1.59 higher chance of achieving disease control than PD-L1-negative patients (OR 1.59, 95% CI 0.78–3.21). The OR was similar considering single-agent (OR 1.50, 95% CI 0.74–3.03) or combination treatments (OR 1.66, 95% CI 0.2–10.68). (Appendix A) Finally, PD-L1-positive patients have a 29% (HR 0.71, 95% CI 0.52–0.96) lower risk of death and a 28% (HR 0.72, 95% CI 0.55–0.95) lower risk of disease progression than PD-L1-negative patients (Appendix A).

Overall, AEs of any grade were reported in 84% of patients (95% CI 78–89%). AEs of any grade were slightly higher with combined treatment (87%, 95% CI 77–97%) than single-agent therapy (82%, 95% CI 74–90%, *p* = 0.01). G3-G4 AEs occurred in 24% of patients (95% CI 13–34%). A higher rate of G3–G4 AEs was observed in combination treatments (28%, 95% CI 21–35%) than in monotherapies (22%, 95% CI 8–36%) (Figure 3).

### 3.5. Meta-Analysis: First-Line Outcomes

Patients treated in first line have a 68% (95% CI 63–72%) probability of survival at 12 months with combined treatment. The 6-month PFS rate was 28% (95% CI 21–35%), whereas the ORR and DCR were 41% (95% CI 36–46%) and 79% (95% CI 68–89%), respectively. The ORR and DCR were statistically significantly higher in first-line treatment rather than in the second or subsequent lines of treatment with the combination treatment (*p* < 0.0001 and *p* = 0.0112, respectively). However, the 12-month OS and 6-month PFS rates did not differ significantly (*p* = 0.0545 and *p* = 0.1464, respectively) (Appendix A).

### 3.6. Meta-Analysis: Second-Line Treatment

The 12-month OS rate for second- and third-line treatment was 52% (95% CI 44–61%) for patients treated with either anti-PD-1/PD-L1, anti-CTLA-4, or the combination of both. The probability of survival at 12 months was higher with ICI combinations (60%, 95% CI 50–70%), followed by single-agent anti-PD-1/PD-L1 (48%, 95% CI 43–53%), and anti-CTLA-4 (40%, 95% CI 27–54%), respectively. The difference was statistically significant (*p* < 0.01). The pooled 6-month PFS estimate was 16% (95% CI 11–21%). This was higher for combination treatments at 25% (95% CI 17–34%), followed by single-agent anti-PD-1/PD-L1 (15%, 95% CI 11–19%) and anti-CTLA-4 (5%, 95% CI 0–10%) treatments. This difference was statistically significant (*p* < 0.01). Taken together, ICI treatments registered a 16% ORR (95% CI, 11–21%) and a 52% DCR (95% CI, 44–61%). The ORR was higher with combination treatment (21%, 95% CI 8–33%) than with single-agent anti-PD-1/PD-L1 (17%, 95% CI 12–22%) and anti-CTLA-4 (5%, 95% CI 2–9%) treatments. The overall DCR was 52% (95% CI, 44–61%). It was higher with combination treatment at 60% (95% CI 50–69%), followed by anti-PD-1/PD-L1 (55%, 95% CI 46–63%) and single-agent anti-CTLA 4 (35%, 95% CI 21–48%) Appendix A.

## 4. Discussion

We performed a systematic review and meta-analysis to assess the impact of immunotherapy, anti-PD-1/PD-L1 and anti-CTLA-4 with or without chemotherapy, in PM patients in terms of PFS, OS, ORR, DCR, and safety.

Our data confirmed the activity of ICIs in PM patients in terms of OS, PFS, and response rate in any line of treatment.

ICIs seem to be more active in untreated patients, as the ORR and DCR are higher than in pre-treated patients. However, despite being numerically higher, the PFS and OS rate did not significantly differ if ICIs were administered in the first or subsequent lines of treatment. This is interesting considering the poor prognosis of PM and the lack of reliable second-line therapies, paving the way to speculate on potential sequential strategies. As expected, the combination of anti-CTLA-4 and anti-PD-1 showed higher OS, PFS, ORR, and DCR both in the first- and second-line settings; however, this was tempered by a higher rate of AEs, particularly those of G3–G4. This therapeutic option should be considered, particularly for younger patients with a good performance status, considering the potential higher risk of severe adverse events. Interestingly, in the second-line setting, single-agent anti-PD-1/PD-L1 had a 17% ORR and a 55% DCR, which are slightly higher than the historical data for “standard” second-line chemotherapy (gemcitabine/vinorelbine) and do not significantly differ from the combination treatment (21% ORR and 60% DCR, respectively). This is in line with what has been reported in another meta-analysis by Tagliamento et al. on 13 studies including 888 pre-treated patients, in which the ORR and DCR were 18.1% (95% confidence interval [CI] 13.9–22.8%) and 55.4% (95% CI: 48.1–62.5%), respectively, whereas the median PFS ranged from 2.1 to 5.9 and the OS from 6.7 to 20.9 months [43]. Moreover, also considering other experimental treatments, such as the combination of chemotherapy and antiangiogenics, single-agent anti-PD1 inhibitors can confer benefit for pre-treated patients with PM, with a favorable toxicity profile, as revealed in another meta-analysis by Banna et al. [44].

On the basis of these and our data and considering the lack of reliable therapy in pre-treated PM patients, single-agent ICIs could possibly still represent an option for those patients who cannot be candidates for combined treatments, for example elderly and frail populations; predictive factors are urgently needed to select patients who can have a benefit. A sequential strategy of platinum–pemetrexed chemotherapy and single-agent anti-PD-1/PD-L1 at disease progression should still be taken into account, particularly in epithelioid PM that does not derive a meaningful benefit from the ICI combinations, as suggested by the Checkmate 743 trial [13]; however, lacking a prospective clinical trial exploring a sequential strategy in this PM subgroup may leave this statement as an interesting speculative perspective to be validated in clinical practice.

Ab anti-CTLA-4 monotherapy was also confirmed for a lack of activity in our meta-analysis, leaving its application for further prospective trials with other agents with a different mechanism of action.

Our meta-analysis has several limitations. First of all, since most of the included studies lacked a comparison arm, we were not able to compare ICIs with single-agent chemotherapy nor to carry out a network meta-analysis. In addition, we should acknowledge that the DOR outcome could not be synthetized in a meta-analysis of published data since its confidence interval is asymmetric and could not be considered normally distributed. The selected trials have used different criteria for response evaluation such as mRECIST for PM and iRECIST and some of the considered cases did not have independent assessment. Furthermore, we could not evaluate the activity of ICIs in pre-treated patients according to histology since a few studies reported specific data.

The results of our meta-analysis suggest a role of ICI combinations in a first-line setting showing a more robust activity and efficacy than in pre-treated populations in which the use of immunotherapy could be reserved for a frail PM subpopulation. However, regulatory concerns in many countries may limit their use. A “financial toxicity” should also be taken into account, particularly for ICI combinations. Cost-effective analysis on output metrics including the patient’s lifetime quality-adjusted life years (QALYs), lifetime costs, and incremental cost-effectiveness ratio (ICER) in United States (US) patients suggest that the combination of nivolumab and ipilimumab may not be a cost-effective choice as an up-front treatment [45,46]. Although the early use of ICIs in the first-line treatment has proved beneficial in other cancers, it is not possible to conclude that this will be the case in all patients with PM. Identification of predictive factor of response may help to select patients who can really benefit and address treatment.

The lack of predictors of response and the confounding effects between predictive and prognostic factors make the issue of immunotherapy efficacy remain confusing.

Our data showed that PD-L1-positive patients have higher probability of response or achieving disease control with ICI treatment than PD-L1-negative patients, considering 1% of expression as cut-off for positivity. This is in line with what was reported by Tagliamento et al. [43]. However, the role of PD-L1 tumor expression is still debated. A recent meta-analysis on this topic performed on 29 trials concluded that PD-L1 status was not an established prognostic nor predictive biomarker [47]. In many trials employing chemotherapy agents, patients with PD-L1 >1% had a higher risk of death compared to their PD-L1-negative counterparts, with a proportional association to the degree of expression [48]. With the introduction of ICIs, some clinical trials demonstrated a trend of favorable effect leading to a longer survival rate in patients with PD-L1-positive tumors than in patients with PD-L1-negative tumors [48]. The small size of the studies analyzed, the heterogeneity of other clinical variables (such as histology, PS, and line of treatment), different PD-L1 assays and clones, and different cut-off points are all factors that negatively influence a definitive conclusion about the role of this biomarker. Curiously, in our meta-analysis the OR was higher for single agents than for combined ICIs. These data may suggest that combining anti-PD-1/PD-L1 with anti-CTLA-4 should increase the chance of a response in PD-L1-negative patients.

Many other confounding effects can determine the uncertainty of the use of ICIs in an unselected population. This is the case for the loss of expression of BAP1, which emerges as a predictor of response to chemotherapy as well as being a possible candidate biomarker for the use of immunotherapy [49,50]. The lack of stratification of the loss of BAP1 expression in the studies analyzed in this meta-analysis, reveals how other predictors of response may cause the final efficacy results of immunotherapy to be unbalanced. Differences in the tumor microenvironment (TME) could be another important confounding factor. PM is considered to have a highly inflammatory TME, as the consequence of an inflammatory response to asbestos exposure [51,52,53]. However, a deeper characterization of its TME revealed the prevalence of chemokines and suppressive immune cells, M2-like macrophages and regulatory T cells, a low tumor mutational burden, and a paucity of activated T cells, which makes PM non-immunogenic [17,18]. Thus, the understanding of the crosstalk and interactions of immune, stromal, and tumor cells is of major importance for the development of novel therapies and the prediction of responses to ICIs.

Many other agents and new therapeutic approaches are under investigation, such as adoptive immunotherapy or vaccines, alone or in combination with ICIs. Preliminary results of a phase I study of intra-pleural injection of mesothelin-targeted chimeric antigen receptor T-cell (CAR-T) therapy, with or without anti-PD-1 agents, showed a 63% RR in 18 MM patients, 37% of which had already been treated with three or more lines of therapy [54]. Results from an ongoing phase III trial of chemotherapy–immunotherapy combination in the first-line setting, along with the results of the Checkmate 743 trial, are expected to change the treatment landscape in the near future. In this regard, we did not include in our analysis the phase II PrE0505 trial, as it was published after the cut-off date for our analysis [55]. This trial demonstrated that durvalumab plus standard chemotherapy improved OS compared with historical control (20.4 vs. 12.1 months). [55] The magnitude of the pooled results did not substantially change with the inclusion of the PrE0505 study [55], although the 12-month OS rate (70.4%), 6-month PFS rate (67.3%), ORR (57.4%), and DCR (94.4%) were higher than the corresponding anti-PD-1/PD-L1 subgroup meta-analytic estimates. Interestingly, integrated genomic and immune cell repertoire analyses in this trial revealed that a higher immunogenic mutation burden coupled with a more diverse T-cell repertoire was linked to a favorable clinical outcome, as well as a higher degree of genomic instability and germline alterations in genes involved in DNA repair [55]. Further analyses to these may help in selecting patients with a higher chance of response and help to individualize the treatment.

Furthermore, a recent publication reported that the antigenic potential of PM could be better predicted by other factors, including chromosomal rearrangements (chromoplexy and chromothripsis) [56]. A recent re-analysis of The Cancer Genome Atlas (TCGA) data classified human tumors based on Tumor Immunogenicity Score (TIGS) as a measure of combined tumor antigen load (e.g., TMB) and antigen presentation capacity. In this analysis, Wang et al. showed that TIGS is a better predictor of the overall response rate to ICI therapy compared with TMB as a single predictive factor [57].

Such interesting data deserve more studies in the context of ICI therapy to identify better predictors of response.

## 5. Conclusions

Our data confirmed the activity of both combined and single-agent ICIs in any line of treatment and suggests that anti-PD-(L)1 single agents might be useful in some chemotherapy pre-treated patients. The efficacy of ICIs is independent from the line of treatment, paving the way for sequential strategy speculations. However, the superiority of ICIs in the standard optimal arm is still questionable as no direct comparison can be performed due to the lack of phase III randomized trials. The “financial toxicity” should still be taken into account and, with regard to this, reliable predictive biomarkers are urgently needed to select patients and customize treatment.

## Figures and Tables

**Figure 1 cancers-14-06063-f001:**
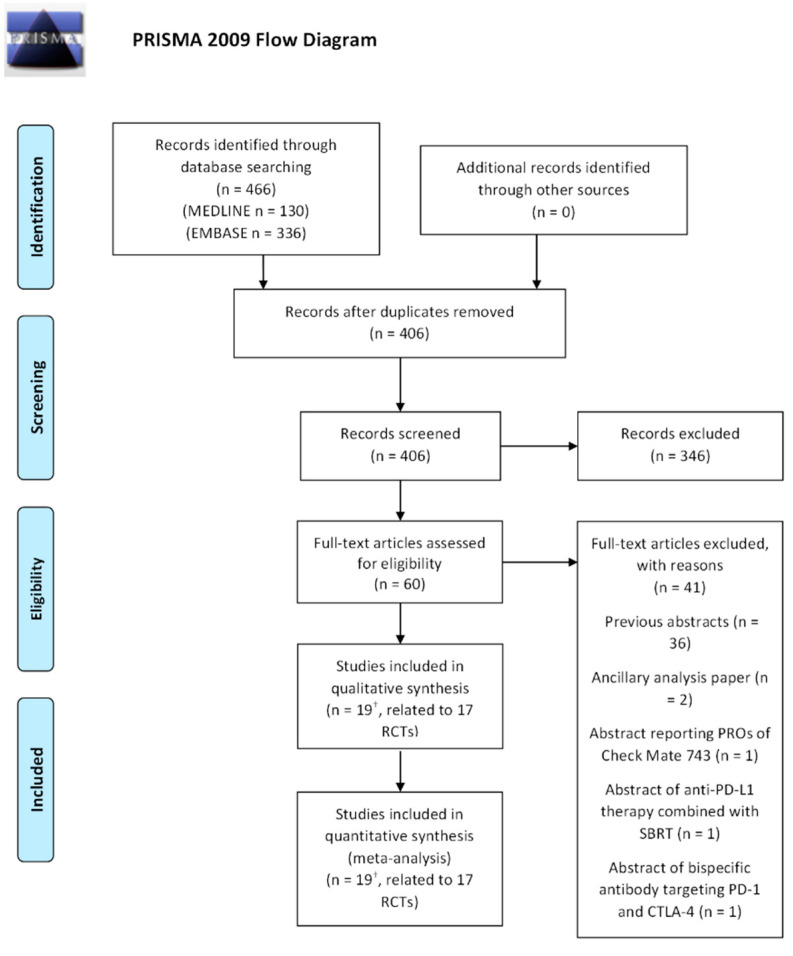
PRISMA flowchart of selected trials. ^†^ The studies by Hayashi et al., 2020 [26] and Calabrò et al., 2021 are updates of the MERIT (Okada et al., 2019) and NIBIT-MESO1 (Calabrò et al., 2018) RCTs, respectively.

**Figure 2 cancers-14-06063-f002:**
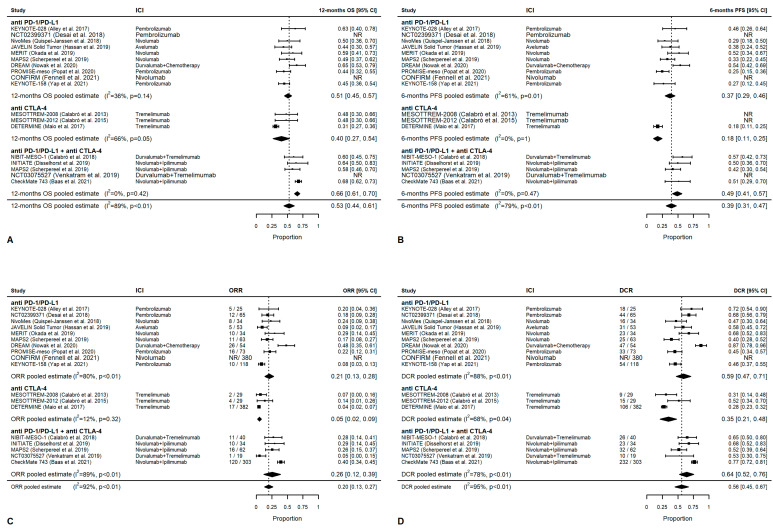
Efficacy endpoint by agent. (**A**) OS probability at 12 months, (**B**) PFS probability at 6 months, (**C**) ORR, and (**D**) DCR.

**Figure 3 cancers-14-06063-f003:**
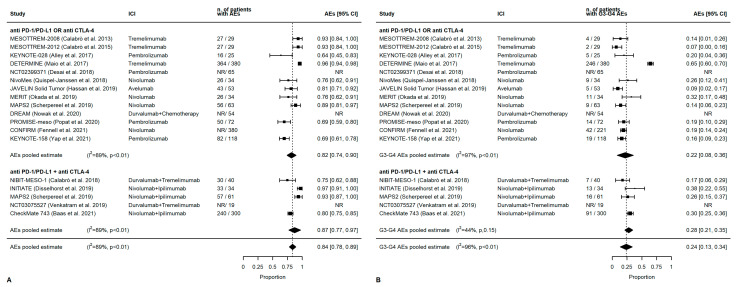
AEs. (**A**) Any grade, (**B**) G3-G4.

**Table 1 cancers-14-06063-t001:** Trial design and primary endpoints. ORR: Overall Response Rate, PFS: Progression Free Survival, OS: Overall Survival, DCR: Disease Control Rate, CA: Centrally Assessed, IA: Investigator Assessed.

Trial	Phase	Blinding	Treatment	Dose	Control Arm	Primary EPs	Stratification Factors
Calabrò et al., 2018, NIBIT-MESO-1 [27,28]	II	open label, single arm	Durvalumab + tremelimumab	Durva: 20 mg/kg 1q28 ipi: 1 mg/kg 1q28	/	ORR	/
Calabro’ et al., 2013 [30]	II	open label, single arm	Tremelimumab	15 mg/kg ev 1q90	/	ORR	EORTC prognostic score
Calabro’ et al., 2015 [31]	II	open label, single arm	Tremelimumab	10 mg/kg 1q28 for 4 cycles, then every 12 weeks	/	ORR	EORTC prognostic score
Nowak A.K et al., 2020, DREAM [29]	II	open label, single arm	Durvalumab + chemotherapy	Durvalumab 1125 mg 1q21 cisplatin 75 mg/m2 or carboplatin AUC5 + pemetrexed 500 mg/m^2^	/	PFS	/
Baas P. et al., 2021, CheckMate 743 [13]	III	open label, randomized	Nivolumab + ipilimumab	Nivolumab 3 mg/kg 1q14 ipilimumab 1 mg/kg 1q42	CT cisplatin or carboplatin + pemetrexed	OS	Sex, Histology
Quispel-Janssen J. et al., 2018, NivoMes [39]	II	open label, single arm	Nivolumab	3 mg/kg 1q14	/	DCR at 12 weeks	/
Popat A. et al., 2020, PROMISE-meso [11]	III	open label, randomised	Pembrolizumab	200 mg 1q21	gemcitabine/vinorelbine	PFS	Histology
Fennel DA et al., 2021, CONFIRM [12]	III	Double blind, randomized	Nivolumab	3 mg/kg 1q14	placebo	OS e PFS (IA, investigator assessed)	Histology
Okada M. et al., 2019, MERIT [38]	II	open label, single arm	Nivolumab	240 mg 1q14	/	ORR (CA, centrally assessed)	/
Maio M. et al., 2017, DETERMINE [32]	IIb	Double blind	Tremelimumab	10 mg/kg 1q28 for 7 doses, then every 12 weeks	placebo	OS	EORTC status, line of therapy, anatomic site
Alley W et al., 2017, KEYNOTE-028 [35]	Ib	open label, single arm	Pembrolizumab	10 mg/kg 1q14 o 1q21 or 2 mg/kg 1q21	/	ORR	/
Yap T. et al., 2021, KEYNOTE-158 [36]	II	open label, single arm	Pembrolizumab	200 mg 1 q21	/	ORR (CA)	/
Hassan R. et al., 2019, JAVELIN [34]	Ib	open label, single arm	Avelumab	10 mg/kg 1q14	/	ORR (IA)	/
Desai et al., 2018 [40]	II	open label, single arm	Pembrolizumab	200 mg 1q21	/	ORR	/
Sherpereel A et al., 2019, MAPS-2 [37]	II	open label, randomized	Nivolumab or Nivlumab + Ipilimumab	Nivolumab 3 mg/kg ipilimumab 1 mg/kg	/	DCR	/
Venkatraman D et al., 2019 [41]	II	open label, single arm	Durvalumab + Tremelimumab	Durval: 1500 mg 1q28 tremel: 75 mg 1q28	/	ORR	/
Disselhorst MJ et al., 2019 INITIATE [33]	II	open label, single arm	Nivolumab + Ipilimumab	Nivo: 240 mg 1q14 ipi: 1 mg/kg 1q42	/	DCR IA 12 W	/

**Table 2 cancers-14-06063-t002:** Results by endpoint. ORR: Overall Response Rate, PFS: Progression Free Survival, OS: Overall Survival, DCR: Disease Control Rate, CA: Centrally Assessed, IA: Investigator Assessed, NE: Not evaluated, NR: Not Reached, CI: confidence interval, IQR: Interquantile Range.

Trial	Pts n (Experimental Arm)	mFUP (Months; IQR)	mPFS (Months) (95%CI)	mOS (Months) (95%CI)	ORR (%) (IQR)	mDOR (Months) (95%CI)	DCR (%) (IQR)
Calabrò et al., 2018, NIBIT-MESO-126, [28]	40	19.2 (13.8–20.5)	8.0 (6.7–9.3)	16.6 (13.1–20.1)	28 (15–44)	16.1 (IQR 11.5–20.5)	65 (48–79)
Calabro’ et al., 2013 [30]	29	27 (23–35)	6.2 (1.3–11.1)	10.7 (0–21.9)	6.9% (0.0–16.1)	12.4 (6–30)	31.0 (14.2–47.9)
Calabro’ et al., 2015 [31]	29	21.3 (18.7–25.9)	6.2 (5.7–6.7)	11.3 (3.4–19.2)	3.4%; 0–10.0)	NE	37.9 (20.2–55.6)
Nowak A.K et al., 2020, DREAM [29]	54	28.2 (26.5–30.2)	6.9 (5.5–9.0)	18.4 (13.1–24.8)	48 (35–61)	5.6 (4.9–12.3)	87 (80–91)
Baas P. et al., 2021, CheckMate 743 [13]	605 (303)	29.7 (26.7–32.9),	6.8 (5.6–7.4)	18.1 (16.9–22.0)	40 (34.1–45.4)	11 (1.45–3.27)	77 (71.4–81.2)
Quispel-Janssen J. et al., 2018 NivoMes [39]	34	27.5 (19.3–NR)	2.6 (2.23–5.49)	11.8 (9.7–15.7)	24 (NR)	7.0 (>3)	47 (NR)
Popat A. et al., 2020, PROMISE-meso [11]	144 (73)	17.5 (9.9–14.5)	2.5 (2.1–4.2)	10.7 (7.6–15.0)	22 (13–33)	4.6 (2.1–NR)	45 (39–55)
Fennel DA et al., 2021, CONFIRM [12]	332 (221)	11.6 (7.2–16.8)	3.0 (2.8–4.1)	10.2 (8.5–12.1)	11 (NR)	4.6 (3.0–6.9)	12 (NR)
Okada M. et al., 2019, MERIT [38]	34	16.8 (1.8–20.2)	6.1 (2.9–9.9)	17.3 (11.5–NR)	29 (16.8–46.2)	11.1 (3.5–16.2)	68 (50.8–80.9)
Maio M. et al., 2017, DETERMINE [32]	571 (382)	NE	NE	7.7 (6.8–8.9)	4.5 (2.6–7.0)	4.8 (26–8.3)	27.7 (16.0–28.3)
Alley W et al., 2017, KEYNOTE-028 [35]	25	18.7 (10.4–24.0)	5.4(3.4–7.5)	18 (9.4-NR)	20 (6.8–40.7)	12.0 (3.7-NR)	72 (NE)
Yap T. et al., 2021, KEYNOTE-158 [36]	118	38.5 (37.5–39.2)	2.1 (2.1–3.9).	10.0 (7.6–13.4)	8 (4–15)	14.3 (4.0–33.9)	46 (NE)
Hassan R. et al., 2019, JAVELIN [34]	53	24.8 (16.8–27.8)	4.1 (1.4–6.2)	10.7 (6.4–20.2)	19 (3.1–20.7)	15.2 (11.1–NR)	58
Desai et al., 2018 [40]	65	NR	4.5 (2.3–6.2)	11.5 (7.6–14)	19	NE	66
Sherpereel A et al., 2019, MAPS-2 [37] (nivo cohort)	125 (63)	20.1 (19.6–20.3)	4.1 (2.8–5.7)	11.9 (6.7–17.7)	19 (8–29)	7.4 (4.1–11.9)	40 (28–52)
Sherpereel A et al., 2019, MAPS-231 (nivo-ipi cohort) [37]	125 (62)	20.1 (19.6–20.3)	5.6 (3.1–8.3)	15.9 (10.7-NR)	28 (16–40)	8.3 (3.0–14.0)	52 (39–64)
Venkatraman D et al., 2019 [41]	19	7.1 (NE)	2.8 (2.0–5.7)	7.8 (6.2-NR)	5 (NE)	NE	52.6 (NE)
Disselhorst MJ et al., 2019 INITIATE [33]	35	14.3 (12.7–15.7)	6.2 (4.1–NR)	NR (12.7-NR)	29 (NE)	14.3 (6.4–NR)	68 (NE)

## Data Availability

Not applicable.

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
