# Peer review of "Immune Checkpoint Inhibitors in Malignant Pleural Mesothelioma: A Systematic Review and Meta-Analysis"

_cancers, 2022, doi:10.3390/cancers14246063_

Round 1
Reviewer 1 Report
This systematic review and meta-analysis investigates the role of ICIs in pleural mesothelioma focusing on the possible role of combined antiCTLA4 + antiPD1/PD-L1 and single agent antiPD1/PD-L1.
In my opinion the most important result of this work from a clinical point of view is that efficacy (ie OS and PFS) seems independent from treatment line (not significantly difference in 12 months OS and PFS in first vs subsequent line) paving the way to relevant consideration on sequential treatments. This point should be better underlined in the summary and abstract and also in the conclusions deserves wider discussion.
Premises and methods are well detailed and reported, including the objectives of the study, the statistical plan and the risk of biases.
What should be deeply revised is the section concerning the results. The most relevant discordance is in the numbering of the reference. There are many mistakes that makes the reading very difficult and I must admit that it took to me really much time checking the references.
As an example see page 4 line 148- 151. Reference 13 is reported both among the 4 trial in pretreated patients and in the 2 trials in naive patients and this is a mistake. There is a clear discordance in the references numbering in the tables and in the text. I started to correct the numbers but there is too much confusion and you have to revise carefully the whole section of the results paying attention that the reference in the text and in the tables are the same and corresponding to the references reported in the bibliography.
I could make many other examples like on page 9 line 22 INITIATE is referenced with 27 and in line 233 the same reference 27 is reported for KEYNOTE-158 and many others mistake like these are present in this section.
Regarding the discussion I suggest to start this section directly with the relevant findings of this work. Indeed in the introduction you already reported the unmet needs and open questions in this highly difficult to treat disease and there is no reason for repeating them in the discussion. On the contrary I would comment on how your results can guide the clinical practice. One of the most important point is the superimposable activity in first and second/further line that to my knowledge wasn't reported in previous analysis. This is highly important for patients and clinicians.
Some statements need to be better explained. Ie on page 13 lines 397 - 400 you state that data of CM743 suggest a sequential strategy but I really cannot understand from which data this consideration came from.
As last consideration, since the 2021 WHO classification no longer includes the term malignant pleural mesothelioma but only pleural mesothelioma, I would suggest to use the new terminology according to the last WHO classification.
Author Response
We thank the reviewer for the comments and suggestions, that allowed us to further enhance the positive results of our study.
We have further evidenced that ICIs efficacy is indipendent from the line of treatment in abstract, summary and results. The discussion has been widely revised, with a focus on this point.
Refrence list has been checked and corrected, particularly in teh text and tables.
As reported above, we have widely modified the discussion part, starting directly with the relevant findings and expanding clinical considerations;
The statement on page 13 line 397-400 has been deleted, a sit was confounding;
The term malignant pleural mesothelioma was replaced by pleural mesothelioma in the text.

Reviewer 2 Report
The authors have submitted a MS providing a systematic review and meta-analysis of clinical trials testing single or combo ICis tested on 2328 MPM. The authors provide evidence that both OS and PFS rate were significantly higher with combo than with single treatment (p 0.001). 12 month OS and 6 month PFS rate did not differ significantly among pre-treated or untreated patients. As expected, the authors show that combined ICIs treatments have significantly higher rate of AEs (p=0.01) and conclude that combined ICIs are more effective but also significantly more toxic than single agents .
The authors have provided a well conducted review and meta-analysis of the current results with ICi for MPM although the paucity of phase III RCT and the choice of suboptimal control arms ends up to affects the clinical trials analysed and, in turn, this study itself
That said the authors should provide more insight into the several critical studies on the efficacy of ICis for MPM during this year :
Meirson T, Pentimalli F, Cerza F, Baglio G, Gray SG, Correale P, Krstic-Demonacos M, Markel G, Giordano A, Bomze D, Mutti L. Comparison of 3 Randomized Clinical Trials of Frontline Therapies for Malignant Pleural Mesothelioma. JAMA Netw Open. 2022 Mar 1;5(3):e221490. doi: 10.1001/jamanetworkopen.2022.1490. PMID: 35262715; PMCID: PMC8908075.
Messori A, Trippoli S. Current treatments for inoperable mesothelioma: indirect comparisons based on individual patient data reconstructed retrospectively from 4 trials. J Chemother. 2022 Apr 12:1-5. doi: 10.1080/1120009X.2022.2061183. Epub ahead of print. PMID: 35411826
Kerrigan K, Jo Y, Chipman J, Haaland B, Puri S, Akerley W, Patel S. A Real-World Analysis of the Use of Systemic Therapy in Malignant Pleural Mesothelioma and the Differential Impacts on Overall Survival by Practice Pattern. JTO Clin Res Rep. 2022 Jan 12;3(3):100280. doi: 10.1016/j.jtocrr.2022.100280. PMID: 35243411; PMCID: PMC8861643.
Yang L, Cao X, Li N, Zheng B, Liu M, Cai H. Cost-effectiveness analysis of nivolumab plus ipilimumab versus chemotherapy as the first-line treatment for unresectable malignant pleural mesothelioma. Ther Adv Med Oncol. 2022 Aug 3;14:17588359221116604. doi: 10.1177/17588359221116604. PMID: 35958872; PMCID: PMC9358333.
Ye ZM, Tang ZQ, Xu Z, Zhou Q, Li H. Cost-effectiveness of nivolumab plus ipilimumab as first-line treatment for American patients with unresectable malignant pleural mesothelioma. Front Public Health. 2022 Jul 22;10:947375. doi: 10.3389/fpubh.2022.947375. PMID: 35937220; PMCID: PMC9354521.
Likewise several critical comments have been raised pointing out the limitations of the multiple Phase II studies for MPM recently published and the authors should deepen their literature search and also provide an evenhanded report of these critical comments
More in general terms this study starts from the assumption that it is taken for granted that the trials with ICis for MPM really top the results of the best control arm and that these trials’ design is flawless. Unfortunately it seems is not the case and the authors should taken account of all the sources prior to getting across an incomplete or even distorted message
On the other hand one cannot do else but appreciating how the authors underscore the risk of toxicity of the combined treatments with ICIs
Last not least the scattered approval of ICIs for MPM by regulatory bodies provides a further evidence of how this treatments’ efficacy is still under scrutiny and their superiority on optimal standard treatments far from being set in stone.
Author Response
We thank the reviewer for the revision and the usefull comments.
We thank also for the references suggested, that we added to the text to improve the introduction and background. The first three are citeted in the introduction, focusing in particular on the CM 743 and the indirect comparisons with chemotherapy and the relative doubts about the use of the combination in the first line setting. The last two references are cited in the discussion, about the cost-effectiveness and financial toxicity of immunotherapy;
The introduction has been further implemented to better explain the background of the study;
About the english, the paper was firstly revised by a mother tongue, as reported in the aknowledgements, and revised a second time after this revision;
We extensively revised the introduction and discussion part to better explain the message of our study on the efficacy of ICis in mesothelioma, with a deeper critical review of the literature and clinical considerations, also about regulatory concerns.
Round 2
Reviewer 1 Report
check the following pints:
1) page 1 - lines 24 and 24 : ORR and DCR values refer to immunotherapy approaches overall? to single agent anti PDL1/PD1 or to combined antiCTLA4 and antiPD1: please clarify
2) page 2 line 67: benvacizumab to be amended
3) page 2 lines 71-73: ORR and DCR values refer to? vino? gem? both? which is the exact reference? Please clarify
4)page 5 lines 194-198: You mention 5 RCT: 4 in pretreated and 2 in naive. This statement is confounding: please amend this sentence
5)page 9 line 241: DREAM ph II has been published 2 years ago, please amend the reference PMID: 32888453
6)page 9 lines 240-246: why you have not mentioned also the ph II with Durva + standard chemo by Forde - PrE0505 - PMID: 34750557 - I suggest to report also on this study
7) page 10 line 308: what does CTLA mean? I don't understand
8) page 13 line 386: i suggest to move "respectively" at the end of the sentence
9) page 14 line 445 : MPM has to be amended in PM
There are many typos in the text that have to be carefully revised
Author Response
We thank the reviewer for these revision and comments. Here we report the response point-by-point
1) the sentence was clarified at page 1 lines 24-26.
2) benvacizumab has been corrected
3) The sentence has been clarified at page 2 lines 63-64 and 71-74
4) The sentence has been amended, as trial ref 36 enrolled both patients.
5) The reference 28 refers to the published study on Lancet Oncology 2020. The phase III is still ongoing.
6) The data of this trial has been published on November 2021, 6 months after the end of our search period for the planned analysis, as declared in PROSPERO registration and in the methods, and for this reason not included in the analysis. However, we added a comment on this point in the discussion.
7) The definition in extenso of CTLA-4 has been provided at page 2 line 75
8) Done, see line 380-382.
9) MPM has to be amended in PM. Corrected.
The text has been further revised by all the authors and another mother tongue and typos corrected.
Reviewer 2 Report
The authors have provided a revised draft addressing most of the comments raised. The review is now certainly more comprehensive and evenhanded. Notwithstanding a few points need further clarifications
line 79, Ref 20. Concerns have been raised on this trial. Mostly on the end point and patient selection and they are worth of being quoted and commented
line 85, Ref 9,10. The authors should remark that Pembrolizumab in second line did not top single agent chemotherapy
line 88, Ref 12. Likewise Testing Nivolumab vs not just a suboptimal arm but even vs placebo in fit and young patients together with the efficacy itself of the intervention arm have raised paramount ethical and methodological concerns that should be properly quoted and commented
lines 89, 90 . the statement here should be consistent each other and one can figure out to merge them to clarify the message
lines 445 and 449-457 . The authors should make themselves clear that, in the light of the subsequent analysis they correctly reported in the revised draft , the superiority of ICi for MPM on standard optimal arm is still questionable
line 468. This statement should be significantly toned down for the concern raised with regards to line 445 and 449-457
"Conclusions" the evidences on the cost benefit ratio and financial toxicity should be added
Minor comments: the authors use different abbreviations for Malignant Pleural Mesothelioma throughout the text and this can result confounding
Author Response
We thank the reviewer for the comments and suggestions. Here the response point-by-point.
1) line 79, Ref 20. Concerns about the trial have been discussed between lines 71-79.
2) line 85, Ref 9,10. A sentence has been added to underline this at page 3 lines 94-95.
3) line 88, Ref 12. This was commented between lines 95-98 page 3.
4) lines 89, 90. The sentences have been linked (see lines 90-93 pages 2-3).
5) lines 445 and 449-457 . This has been further underlined in the conclusions.
6) line 468. The sentence was modified and toned down; see lines 432-436, page 14.
7) Cloncusions have been modified accordingly.
Typos have been corrected as well as the abbreviations.
Round 3
Reviewer 2 Report
The authors have properly addressed the comments, and amended the MS accordingly